# Step-by-Step Regeneration of Tentacles after Injury in *Anemonia viridis*—Morphological and Structural Cell Analyses

**DOI:** 10.3390/ijms24108860

**Published:** 2023-05-16

**Authors:** Claudia La Corte, Nicolò Baranzini, Mariano Dara, Camilla Bon, Annalisa Grimaldi, Maria Giovanna Parisi, Maria Grazia Zizzo, Matteo Cammarata

**Affiliations:** 1Marine Immunobiology Laboratory, Department of Earth and Marine Sciences (DiSTeM), University of Palermo, Viale delle Scienze, Ed. 16, 90128 Palermo, Italy; claudia.lacorte@unipa.it (C.L.C.); mariano.dara@unipa.it (M.D.);; 2Department of Biotechnology and Life Science, University of Insubria, Via Dunant 3, 21100 Varese, Italy; nicolo.baranzini@uninsubria.it (N.B.); cbon@uninsubria.it (C.B.); 3Department of Chemical and Pharmaceutical Biological Sciences and Technologies (STEBICEF), University of Palermo, Viale delle Scienze, Ed. 16, 90128 Palermo, Italy; mariagrazia.zizzo@unipa.it; 4Advanced Technologies Network (ATeN) Center, Università di Palermo, Viale delle Scienze, 90128 Palermo, Italy

**Keywords:** anthozoan regeneration, *Anemonia viridis*, PET, histology, immune cells, TEM analysis

## Abstract

Benthic marine invertebrates, such as corals, are often subjected to injury caused by several sources. Here, the differences and characteristics in injured and health tissues in terms of cellular components are shown through a histological investigation of the soft coral *Anemonia viridis* at 0 h, 6 h, 24 h, and 7 days after injury caused by tentacle amputation. In addition, a new tool was used for the first time in invertebrates, positron emission tomography, in order to investigate the events that occur during regeneration within a longer time period (0 h, 24 h, and 14 days after the tentacles were cut). Higher integrated density values were measured through a densitometric analysis in sections stained with Fontana–Masson at 24 h after the tentacles were cut. This suggests an increase in melanin-like containing cells and a subsequent increase in fibroblast-like cells differentiated by amoebocytes that converge to the lesion site in the early stages of inflammation and regeneration. This work provides, for the first time, an elucidation of the events that occur during wound-healing and regeneration in basal metazoan, focusing on the characterisation of immune cells and their role. Our results indicate that Mediterranean anthozoan proves to be a valuable model for studying regeneration. Many events highlighted in this research occur in different phyla, suggesting that they are highly conserved.

## 1. Introduction

Several studies have evidenced the global decline in coral populations caused by a variety of disturbances [1] and events that cause wounding such as predation and natant anchorages [2,3]. To date, only a few histological studies have been conducted on the mechanisms involved in the events that occur during wound-healing and regeneration in anthozoans [4,5,6]. This argument is better addressed in scleractinian corals, in particular for the global observation of regeneration [7,8] under different stress conditions. Although the histological aspects of wound healing have not been thoroughly investigated, they represent the first step in understanding cnidarian immune responses, resulting in better insight into the health status of these organisms. Injury could trigger diseases, facilitating microbe infection [3]. Cnidarians have evolved and developed a suite of strategies to face these challenges, such as clotting, cell proliferation, the activation of inflammation pathways, and the production of metabolites such as melanin. Studies have also shown the clear importance of wound healing as a vital process, in which immune cells are recruited to the lesion site, not only facilitating the healing and blocking of eventual infections but also supporting tissue regeneration [5,9]. Several research have demonstrated the conservation of the main step-points of wound-healing process along phyla [10,11,12], with four characteristic phases [13]: (i) During the coagulation process, semisolid haemolymphs form a plug and seal the injury. This phase leads to the formation of a hard clot, which in mammals, is normally constituted by aggregated platelets and stabilised by fibrin molecules while, in invertebrates, occurs via immune cell degranulation, followed by the incorporation of cellular debris, bacteria, and extracellular matrix components into extracellular aggregates [14,15]. (ii) Immune cells are recruited to the site of the injury in order to engulf aggregates containing microbes and cellular debris [15]. (iii) Proliferation of new immune cells, of fibroblasts-like cells and re-epithelisation. During this phase, different cell types migrate to the edge of the wound, immersed in the extracellular matrix, and join together [16]. In particular, the massive proliferation of fibroblast-like cells, characterised by extensive pseudopods essential for wound contraction and by the synthesis and deposition of new collagen, favours subsequent tissue reorganisation. (iv) During the maturation and tissue remodelling phase, the density of fibroblast-like cells decreases through apoptosis and collagen-rich scar tissue is formed, resulting in an increase in tissue stiffness and strength [16,17]. Although many of these processes have been well-described in vertebrates and in some invertebrates, such as annelids and molluscs [6,9,18,19,20,21], this information is still rather incomplete for many other invertebrate species, especially for anthozoan cnidarians. In order to fill this gap, in this work, we used the experimental model *Anemonia viridis* to shed light on the cellular mechanisms underlying immune response and wound healing. *A. viridis* is a Mediterranean hexacoral anthozoan that can be found in low rocky depths. Like many anthozoans, it is involved in trophic mutualistic endosymbiosis, hosting unicellular zooxanthellae algae belonging to the genus *Symbiodinium*. These organisms consist of two body layers derived from the endoderm (gastrodermis) and ectoderm (epidermis) of the embryo. Between these layers, a gelatinous and amorphous connective layer is present, called the mesoglea, which does not contain cells apart from a few amoebocytes [5] (Figure 1).

Through histological and ultrastructural approaches, we morphologically and functionally characterised the different cell types involved both in the first phase of wound closure and in the regeneration and regrowth of the tentacle following its amputation. Furthermore, using positron emission tomography (PET), for the first time, in invertebrates, we demonstrated how the migration and flow of endosymbionts occur during these processes.

## 2. Results

### 2.1. Positron Emission Tomography (PET) Analysis

The control organism showed a generally high epi-fluorescence that extended from the body to the tentacles (Figure 2A). On the contrary, 1 h after amputation, the signal disappeared in the area that was cut (Figure 2B). Observations 24 h after the tentacles were cut showed fluorescence re-establishment in the whole animal, likely an index of recovery by the endosymbiont in the injured area (Figure 2C). Total recovery of the signal was observed at 14 days, reaching the same fluorescence levels as the control organism, evidence of the restoration of natural conditions (Figure 2D).

### 2.2. Cell Measurements

The agranular amoebocytes from the biometric analysis had a mean area of 19.58 ± 14.42 µm^2^. Their mean length was 8.89 ± 3.60 µm. Their nuclei occupied 40.86 ± 16.01% of the cellular space. The zooxanthellae symbionts had a diameter of 8.27 ± 0.26 µm and a mean area of 53.06 ± 1.74 µm^2^. The granulocytes had a highly variability in size, a mean length of 10.99 ± 3.53 µm, and a calculated mean area of 42.27 ± 23.49 µm^2^. Finally, the mucocytes had an average length equal to 38.67 ± 4.76 µm and a mean area of 282.83 ± 73.44 µm^2^.

### 2.3. Morphological Analyses and Characterisation of Cells Involved in Tentacle Regrowth in Injured A. viridis

In order to shed light on the cell types involved in wound healing and tentacle tissue regeneration from injured *A. viridis*, morphological analyses were performed, both via light microscopy and TEM on the just amputated area (T0) and on the regenerating tentacle 6 h, 24 h, and 7 days after cutting. The results were compared with tissue from a healthy tentacle.

#### 2.3.1. Healthy Tentacles

In the healthy animal control, the tentacle was about 2.5 up to 3 mm long and its epidermis consisted of a single layer of columnar cells arranged obliquely to the main axis of the tentacle and with the basal part lying on the mesoglea layer (Figure 3A). In the part facing outwards, these cells were ciliated and were joined via obvious junctions (Figure 3B). Proceeding towards the innermost region, the epithelial cells appeared densely compacted and contained numerous symbiotic zoochlorellae algae (Figure 3A,C). The epidermis was also populated in the innermost portion by spirocysts; mucus-cell-containing electron-lucent granules; and gland cells with electron-dense spherical or ovoid granules, with diameters varying from 1 to 2.5 µm (Figure 3C,D). The electron density of the granules was generally homogeneous; however, some granules possessed a small, more electron-dense central core (Figure 3D). Interestingly, these glands containing heterogeneous electron-dense granules observed in *A. viridis* appeared similar to serous gland cells secreting lysozymes described for mammals and was considered a defence system against infections [22].

The mesoglea was visible between the epidermis and gastrodermis. It consisted of an amorphous fibrous matrix (Figure 3A) in which a few roundish amoebocytes with large nuclei were detectable (Figure 3E). The monolayered gastrodermis, containing zymogen-type gland cells [23], was formed by columnar cells, including numerous symbiotic zoochlorellae and larger endosymbiotic dinoflagellate zooxanthellae specific to this epithelial layer (Figure 3A,F).

#### 2.3.2. Tentacle Amputation at Time Zero

Immediately post-injury, the epidermis and the gastrodermis of the severed tentacles were destroyed, the cell outlines were no longer distinguishable, and the symbiotic algae were extruded from the epithelial cells (Figure 4A–C). Gland cells, with empty granules, were observed in the cut region (Figure 4B). Waste material, including spirocysts, gland cells, and symbiotic algae, was abundant in the area adjacent to the cut (Figure 4D).

#### 2.3.3. Six Hours after Tentacle Amputation

Six hours after the injury, the regenerating tentacle buds measured approximately 1.8 mm in length (Figure 5A). The thick layer of a mucous substance deposited on the cut surface (Figure 5B) suggested its important role in forming the first barrier against infection. Indeed, the mucus layer provided a physical shield preventing bacteria and debris from accumulating on the regrowing tentacle surface. Large degranulating mucus-secreting and gland cells were visible just below the injured area (Figure 5C). The innermost part of the new bud was populated by both granular and agranular amoebocytes, with the latter containing small, electron-dense granules (Figure 5D). Granular (Figure 5E) and agranular (Figure 5F) amoebocytes were visible in the mesoglea and infiltrated the regrowing bud [5], migrating towards the injured area (Figure 5G).

#### 2.3.4. Twenty-Four Hours after Tentacle Amputation

The regrown tentacles measured about 2 mm in length (Figure 6A). Granular amoebocytes migrating from the mesoglea had reached the injured area and formed a thin layer along the lesion edge (Figure 6B), establishing an epithelial front and contributing to the formation of a plug. This clot formation was similar to what was previously observed for other anthozoans [24], where melanin-containing granular cells were reported. Inside this newly formed epithelial-like layer was numerous mucous and secreting gland cells with empty granules (Figure 6B). The presence of both mucous and empty-appearing gland cells confirmed their possible role in secreting many active products, including mucins and lysozymes, as a mechanism of defence against the entrance of foreign organisms. Dense aggregations of agranular amoebocytes were also present throughout the wound-healing site (Figure 6C).

#### 2.3.5. Seven Days after Tentacle Amputation

Seven days after the cut, the regenerated tentacles were about 2.5–3 mm long, and their shape and morphology were very similar to the control tentacles (Figure 6D). At the superficial level, cellular cilia and intercellular joint points were visible (Figure 6E). The outer layer of granular amoebocytes was no longer present, and the newly formed epidermis mainly consisted of differentiating epithelial cells containing the symbiotic zoochlorella (Figure 6F), and numerous gland and mucous cells (Figure 6G). Well-differentiated columnar cells containing the symbiotic algae formed the gastrodermal layer (Figure 6H).

### 2.4. Colorimetric Analyses of Cells Involved in Wound Healing

Mucicarmine staining (Figure 7A–E), specific for highlighting the presence of epithelial acidic mucopolysaccharides (mucins), and Fontana–Masson staining (Figure 7F–J), specific for highlighting melanin, were carried out on histological sections of both the control and regrown tentacles after different time periods, since their amputation for the regrowth case. The obtained results clearly showed the different roles of the cells involved in wound closure. Indeed, while in the CNT tentacle, positivity for mucicarmine (Figure 7A) and Fontana–Masson (Figure 7B) was very low or absent, after cutting, increasing positivity was observed for both stains, but in reverse order. Immediately after tentacle amputation (T0), a strong, intensely fuchsia-coloured signal was localised in the mucous cells at the edge of the lesion (Figure 7B). After 6 h, the signal was also visible on the external surface, as a result of mucins secretion by the mucous cells (Figure 7C). The mucus secretion decreased after 24 h (Figure 7D) and disappeared completely after 7 days, where the positivity remained confined to only within the mucous cells (Figure 7E).

Conversely, melanin positivity was low at times T0 (Figure 7G) and 6 h (Figure 7H). It was mainly confined to the few granular amoebocytes previously described (Figure 6D–G), localised in the basal region of the epidermis bordering the mesoglea and migrating towards the injured area.

Twenty-four hours after amputation (Figure 7H), the signal for Fontana–Masson was very dark and localised in the superficial area corresponding to the layer of granular amoebocytes involved in the formation of the plug. At 7 days post-injury, the melanin signal was absent (Figure 7J), in agreement with the absence of granular amoebocytes on the newly formed epithelial layer, similarly to that in the CNT.

The densitometric analysis (Figure 8) calculated on tentacle sections at various times from amputation and stained with Fontana–Masson was represented in a graph and confirmed the histological data.

## 3. Discussion

Cnidarians are exposed to physical damage from a variety of sources that injure tissues and result in open wounds. The aim of this research was to explore and characterise, for the first time, the cells present during the wound-healing process and their functions in the soft coral *A. viridis*.

As demonstrated in our previous work focused on this anthozoan, lesions are completely healed within 24 h [5], as already observed in other corals such as *Porites cylindrica* [25], and the wound-healing process follows all of the four previously characterised phases. In particular, granulocytes, immune cells already well-described in the literature [26,27,28], degranulate and release antimicrobial and cytotoxic molecules, including peroxidase, lysozyme, phenoloxidase, and melanin. These cells have been observed in other cnidarians and are involved in pathogen removal at the lesioned site before plug formation occurs [16,29,30].

As shown in Figure 3 and Figure 5, disrupted cells, degranulated granulocytes, and possible pathogens are immediately expelled when an injury arises. Likewise, the endosymbiont algae appear to not be functional at the time when a cut occurs, suggesting that it incurs the same fate.

A similar behaviour during regeneration was also observed in *Calliactis parasitica,* where the different roles of the cells and the interactions of the ectodermal and mesoglea layers have been investigated [4].

In contrast, six hours post-injury (Figure 3), in *A. viridis*, eosinophilic (acidophilic) amoebocytes densely gather, as evidenced by the intense pink colour derived from H&E staining. The high density of these cells in the injured area is explained by their role as effectors of an immune response to infection, contrariwise to the state of healthy tissue, in which they are quite completely absent [31,32]. A similar result was detected in the study by Mydlarz et al., (2008) conducted on *Gorgonia ventalina,* in which animals were exposed to the fungal pathogen *Aspergillus sydowii* [27]. Indeed, a massive and dense increase in amoebocytes was evident in the infected coral tissues. Moreover, in the Caribbean soft coral *Plexaurella fusifera*, after three weeks of healing, the authors not only observed the four major stages of wound healing but also underlined the key role of these cells in the re-epithelisation of the injured area [6]. In particular, acidophilic granular amoebocytes promptly constituted an epithelial-like layer by aggregating and spreading along the injury’s edge. Interestingly, similar findings were also found in other invertebrates [4,33].

Moreover, the evidence obtained in this work suggests that the granules of these cells might contain melanin, which could be necessary to strengthen the epidermal layer during re-epithelialisation, as already described in the literature [25]. However, further molecular studies could be necessary to support this hypothesis.

At the same time, post-injury, oval hyaline cells, also involved in activation with few or no granules, have been detected migrating to the wound area. Compared with granular amoebocytes, these cells possess a less dense cytoplasm, and both the size and the nucleus are more similar to those of fibroblasts or granulocytes. This may suggest that cells originate from a common cell type that differentiate during various stages of wound healing and regeneration.

The differences observed after Masson–Fontana staining, relative to *A. viridis*’s structural organisation, are clearly evident when comparing the tissues after 24 h and 7 days from injury. This could be consistent with the reorganisation activity of fibroblasts induced by collagen deposition, as already demonstrated (see [5]). Our evidence regarding the high presence of melanin at 24 h after amputation of the tentacles appears to be consistent with what was observed in *P. cylindrica* [25], suggesting structural support during regeneration closely related to the increase in fibroblasts-like cells. In addition, a defensive role could be attributable to mucous and empty-appearing gland cells secreting several active compounds, such as mucins and lysozyme (Figure 4). As revealed by the PET observation, in the whole injured area, no signals were registered by the device 1 h post tentacle amputation. These data are consistent with the loss of functionality by symbionts, also visible at the histological level. Moreover, the initial loss of the symbionts, derived from their expulsion from the anemone, could be related to a defence strategy induced by conditions of stress. A similar behaviour was also observed in *P. cylindrica,* in which the effect could be imputed to the cytotoxicity induced by the degranulation of granular cells [21].

Consistent with recovering tissue, *Symbiodinium* cells were again visible between 24 h and 7 days (Figure 2E–G) following lesion reorientation of the cells, and reorganisation of their morphology occurred, observed as lining up side-by-side and perpendicularly to the injury site. Such cell organisation is characteristic of a healthy cnidaria epidermis [34,35,36].

However, the presence of cells rich in melanin-like granules evidenced by Masson–Fontana staining suggests that granular eosinophil amoebocytes could include prophenoloxidase, the precursor to melanin synthesis.

Interestingly, as shown in mucicarmine-stained tissue sections, mucus plays a fundamental role in sealing the injury and in isolating the lesioned site from external potential pathogens. At 24 h and 7 days after injury, mucus is confined to and accumulated as granules to be subsequently expelled by degranulating mucocytes (Figure 3). It could be argued that the apoptotic mechanism is activated, and the dead cell material is first accumulated and then eliminated, thus contributing to the healing process and to proper systemic functioning of the organism [37,38]. Evidence also emphasises the role of mucocytes and mucus secretion under stress, also reporting a change in the density of these cells [39].

## 4. Materials and Methods

### 4.1. Sample Collection

*A. viridis* specimens were collected in autumn 2021 on the coastal area of Palermo (Italy). The animals were housed in facilities at the Marine Immunobiology laboratory (University of Palermo, Italy) and maintained in filtered and oxygenated seawater at 18 ± 1 °C for at least one week before initiating the experimental design.

The experimental groups (N = 30 in duplicate) consisted of three control organisms (healthy organisms) and twelve injured organisms (removal of 25% of the total tentacles), which were observed for histological analyses at different times: 1 h, 6 h, 24 h, and 7 days after amputation. As regards PET technique (N = 12), were collected other three organisms for each time point and for the controls. The time points at which examination occurred were 1 h, 24 h, and 14 days, in order to observe the event over a longer time period.

### 4.2. Positron Emission Tomography (PET)

Epifluorescence in the *A. viridis* total bodies was monitored at different time points (t = 0, 1 h, 24 h, and 14 days) with PET, a nuclear diagnostic imaging technique, in order to clarify the processes following cutting of the tentacles. PET was performed using an IVIS Spectrum system (Perkin-Elmer, Hopkinton, MA, USA) at the animal facility of the ATeN Center, University of Palermo. The epifluorescence of *A. viridis* was imaged using the absorption and emission spectra of the endosymbiont *Symbiodinium* sp. (chlorophyll a, excitation filter at 495 nm, and emission filter at 595 nm). The fluorescent intensity was normalised to a background signal to follow only the zooxanthellae signal.

### 4.3. Light and Transmission Electron Microscopy (TEM)

#### 4.3.1. Embedding Tissue in Paraffin and Staining

The healthy tentacles and the budding area of the regrowing tissues from three *A. viridis* subjected to amputation of 25% of the total number of tentacles at four time points (0 h, 6 h, 24 h, and 7 days after amputation) were fixed in 4% paraformaldehyde in PBS for 2 h, washed several times in PBS solution, dehydrated in increasing concentrations of ethanol (30%, 50%, 70%, 90%, 96%, and absolute), and paraffin-embedded. Sections (7 mm thick) were obtained with a rotary microtome (Jung multicut 2045, Leica, Wetzlar, Germany) and processed for conventional histological staining with haematoxylin and eosin (HE) and for two specific stains, as suggested by the datasheets: Masson–Fontana staining to visualise the melanotic pigment (Masson Fontana kit, Bio Optica, Milan, Italy) and mucicarmine staining to highlight acidic mucopolysaccharides (mucins) of epithelial nature (Mucicarmine kit, Bio Optica). Images were recorded with a Nikon Digital Sight DS-SM Light Microscope (Nikon, Tokyo, Japan).

#### 4.3.2. Embedding Tissue in Epoxy Resin for TEM

*A. viridis* tissues, dissected from healthy tentacles and from regenerating areas, were fixed for 2 h in 4% glutaraldehyde in a cacodylate buffer (pH 7.4). After several washes in the same buffer, the samples were post-fixed for 1 h with 1% osmium tetroxide in a 0.1M cacodylate buffer (pH 7.4). After serial ethanol dehydration (70%, 90%, and 100%), the specimens were embedded in an Epon-Araldite 812 mixture. For the ultrastructural TEM analyses, ultrathin sections (80 nm in thickness), obtained with a Reichert Ultracut S ultratome (Leica), were placed on copper grids (300 mesh, Sigma-Aldrich, Milan, Italy), counterstained with uranyl acetate and lead citrate, and observed with a Jeol 1010 EX transmission electron microscope (Jeol, Tokyo, Japan). Images were recorded with a MORADA digital camera system (Olympus, Tokyo, Japan).

### 4.4. Biometric Measures

The colour intensity of the Fontana–Masson-stained sections was measured as integrated density values (the product of the area and mean grey values of pixels within the selection) using an ImageJ software package on 6 fields (45,000 μm^2^) for each slide [40]. Similarly, on six TEM microphotographs of agranular amoebocytes, granulocytes, mucocytes, gland cells, and zooxanthellae, the length, width, and area of the cell were calculated (see the Appendix A for an example).

### 4.5. Statistical Analyses

The data were tested for ANOVA assumptions, and differences between the groups were shown using a one-way ANOVA as mean ± standard deviation (SD). Differences between the means were considered significant for *p* < 0.05. Tukey’s multiple comparison post hoc test was applied in order to establish significant differences between the control and injured tissue at the different time points using GraphPad Prism 8.0.2 (GraphPad Software, La Jolla, CA, USA).

## 5. Conclusions

In the present work, conducted in *A. viridis*, it has been observed, once again, that the wound-healing process is preserved among metazoans. In fact, this work focused on the events that occur during regeneration and tentacle regrowth, describing all of the intercurrent phases, and our results are applicable to different organisms from various taxonomies. Cells of the immune system have been identified and characterised for the first time, including their distribution, contents, and hypothetical role in the regeneration process.

Future studies are encouraged to explain the events that occur during wound healing and regeneration using molecular tools and immune markers. Furthermore, at this time, this study represents an interesting point of view in comparative morphological analyses, showing a novel zoological scenario.

## Figures and Tables

**Figure 1 ijms-24-08860-f001:**
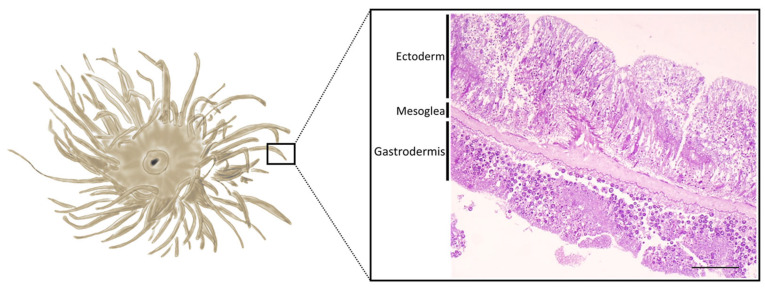
Illustration of the epithelial layers of *Anemonia viridis* tentacles. The outermost layer is the ectoderm and the innermost layer is the gastrodermis, both rich in cells. The latter also hosts zooxanthellae. The mesoglea is the central amorphous gelatinous layer and only has a few cells, predominantly amoebocytes. Scale bar: 100 µm.

**Figure 2 ijms-24-08860-f002:**
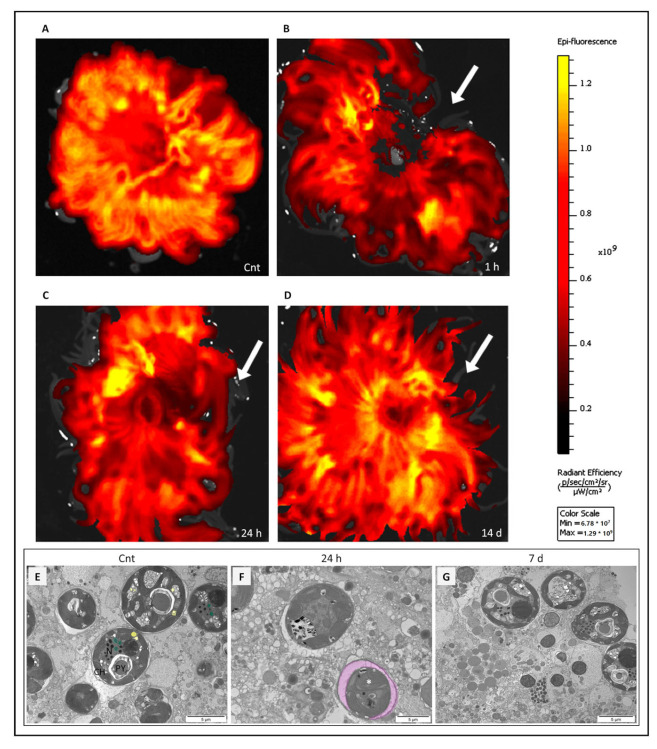
Representative images of in vivo positron emission tomography (PET) of *Anemonia viridis* samples and TEM observation of *Symbiodinium* sp. during wound healing and regeneration. (**A**) Cnt = control organism, not injured, high epi-fluorescence in the whole body; (**B**) 1 h = animal observed one hour after the tentacles were cut, no signal in the wound area (arrow); (**C**) 24 h = *A. viridis* observed 24 h after the tentacles were cut, re-establishment of the signal (arrow); (**D**) 14 d = organism observed 14 days after tentacle amputation, tentacular regeneration occurred. The signal intensity is comparable with that of the control (arrow). (**E**) Symbionts in healthy tissues (control organisms), well-defined cell compartments. N = *Symbiodinium* nucleus; CH = chloroplasts; PY = pyrenoid; in yellow = starch granule; in green = accumulation body. (**F**) Symbionts at 24 h after injury have a lower electron density compared with the control; organelles are not clearly identifiable. In pink = enlargement of membrane, white asterisk = chloroplast swelling. (**G**) At 7 days after the tentacles were cut, *Symbiodinium* sp. appeared to recover their functionalities. Scale bar: 5 µm.

**Figure 3 ijms-24-08860-f003:**
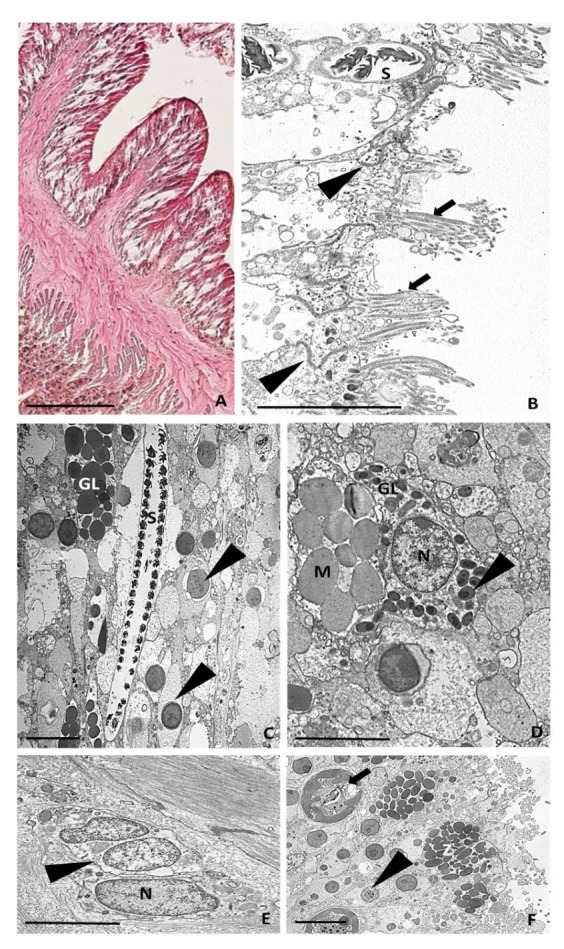
Morphological analyses from light microscope and TEM. (**A**) Image of control tentacle, 2.5 mm long, from light microscope. (**B**) TEM image of apical side of epidermis. Ciliated cells (arrows) joined by junctions (arrowheads) and spirocysts (S) are visible. (**C**) TEM image of innermost part showing columnar cells containing symbiotic zoochlorella algae (arrowheads), spirocysts (S), and gland cells (GL) with electron-dense spherical or ovoid granules. (**D**) Detailed TEM image of mucous cell containing electron-lucent granules (M) and gland cells with large nuclei and containing granules with an electron-dense small central core (arrowheads). (**E**) Detailed TEM image of amoebocytes (arrowheads) with large nuclei (N) localised in the mesoglea. (**F**) Detailed TEM image of the gastrodermal layer. Zymogenic gland cells (Z), zoochlorellae (arrowheads), and zooxanthellae (arrows) are visible. Bar in (**A**), 100 µm; bars in (**B**–**D**,**F**), 5 µm; bar in (**E**), 2 µm.

**Figure 4 ijms-24-08860-f004:**
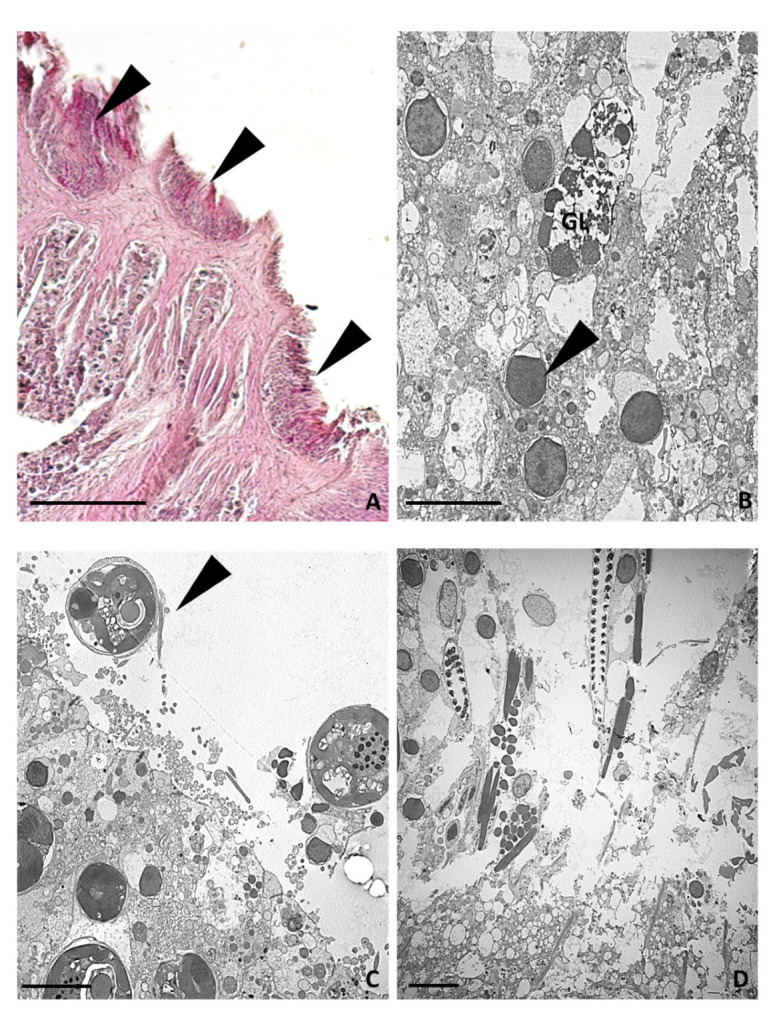
Morphological analysis of the area just amputated (T0) from light and TEM microscopes. (**A**) Overview from light microscope showing the area where the tentacles were cut away. The arrowheads indicate what remains of the tentacle after its amputation from the animal body. (**B**,**C**) TEM images showing the destroyed epidermal (**B**) and gastrodermal (**C**) cell layers from which symbiotic algae (arrowheads) were expelled. A gland cell with empty granules is visible (GL). (**D**) Detailed TEM image of expelled waste material. Bar in (**A**), 100 µm; bars in (**B**–**D**), 5 µm.

**Figure 5 ijms-24-08860-f005:**
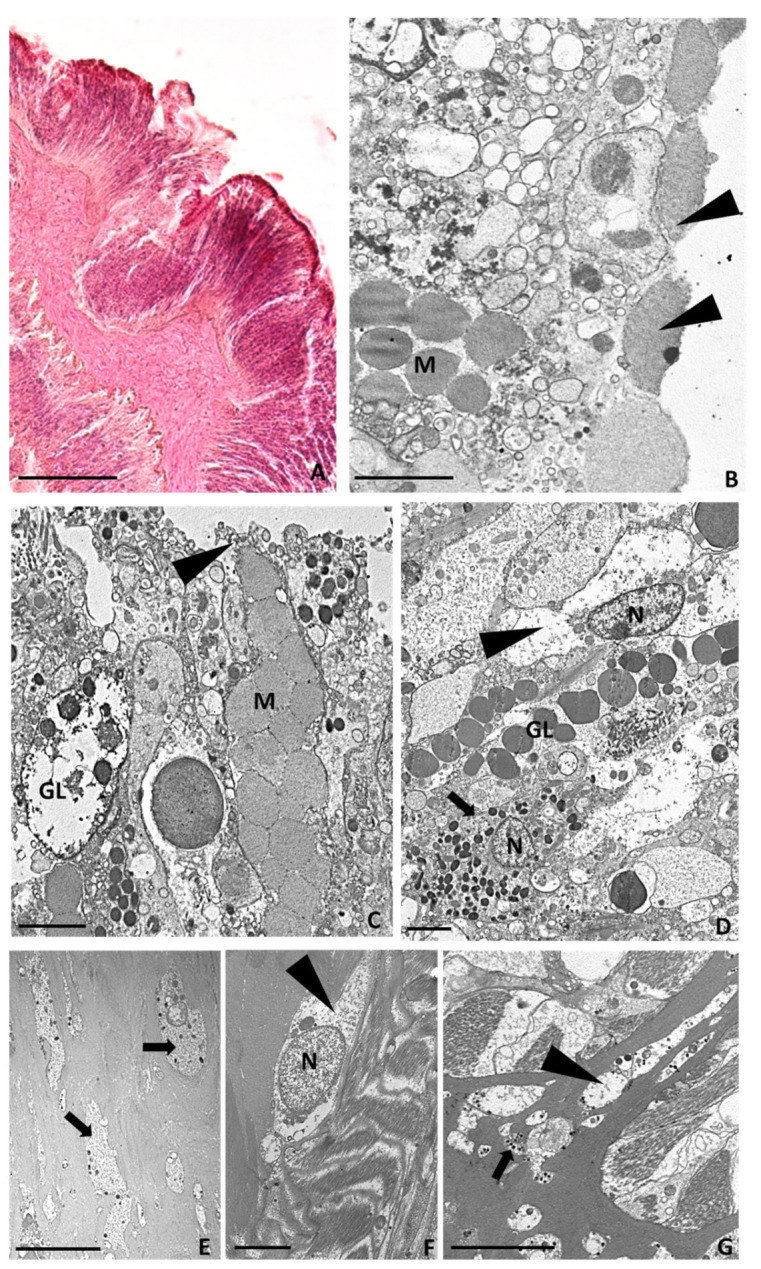
Morphological analysis of the tentacle bud regrown 6 h after cutting from light and TEM microscopes. (**A**) Light microscope overview showing the regrown tentacle bud. (**B**–**G**) Ultrastructural TEM analysis showing mucous cells (M) and gland cells (GL), some of which, with empty granules, are visible just below the cut area. A thick mucus layer (arrowheads in (**B**)) as product of mucous cells (arrowheads in (**C**)) is visible at the edge of the lesion. Agranular (arrowhead in (**D**)) and granular amoebocytes (arrows in (**D**)) are visible in the regenerating epidermal layer. (**E**–**G**) Detailed TEM images of granular (arrows in (**E**,**G**)) and agranular amoebocytes (arrowheads in (**F**,**G**)) present in the mesoglea and migrating towards the wound-healing region. Bar in (**A**), 100 µm; bars in (**B**–**D**,**F**), 2 µm; bar in (**E**,**G**), 5 µm.

**Figure 6 ijms-24-08860-f006:**
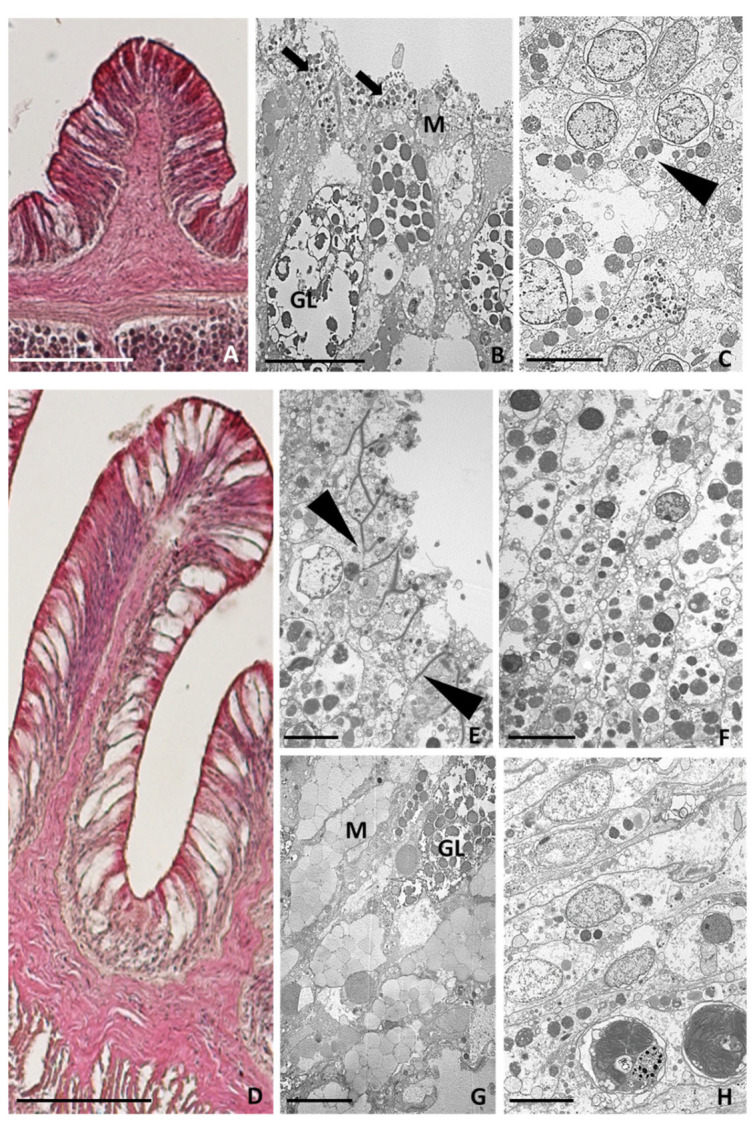
Morphological analysis of the regrown tentacle 24 h (**A**–**C**) and 7 days after amputation (**D**–**H**) via light and TEM microscopes. (**A**) Light microscope overview of 2 mm long regrown tentacle 24 h post-injury. (**B**,**C**) Detailed TEM images showing a layer of granular amoebocytes (arrows in (**B**)) along the newly forming epithelial layer and agranular amoebocytes (arrowheads in (**C**)) containing symbiotic algae. (**D**) Light microscope overview of 3 mm long regrown tentacle 7 days post-injury. (**E**–**H**) TEM images showing intercellular junctions (arrowheads in (**E**)), differentiated epithelial cells in epidermis (**F**) and in gastrodermis (**H**), and gland (GL) and mucous (M) cells (**G**). Bar in (**A**,**D**), 100 µm; bars in (**B**,**C**,**E**–**H**), 5 µm.

**Figure 7 ijms-24-08860-f007:**
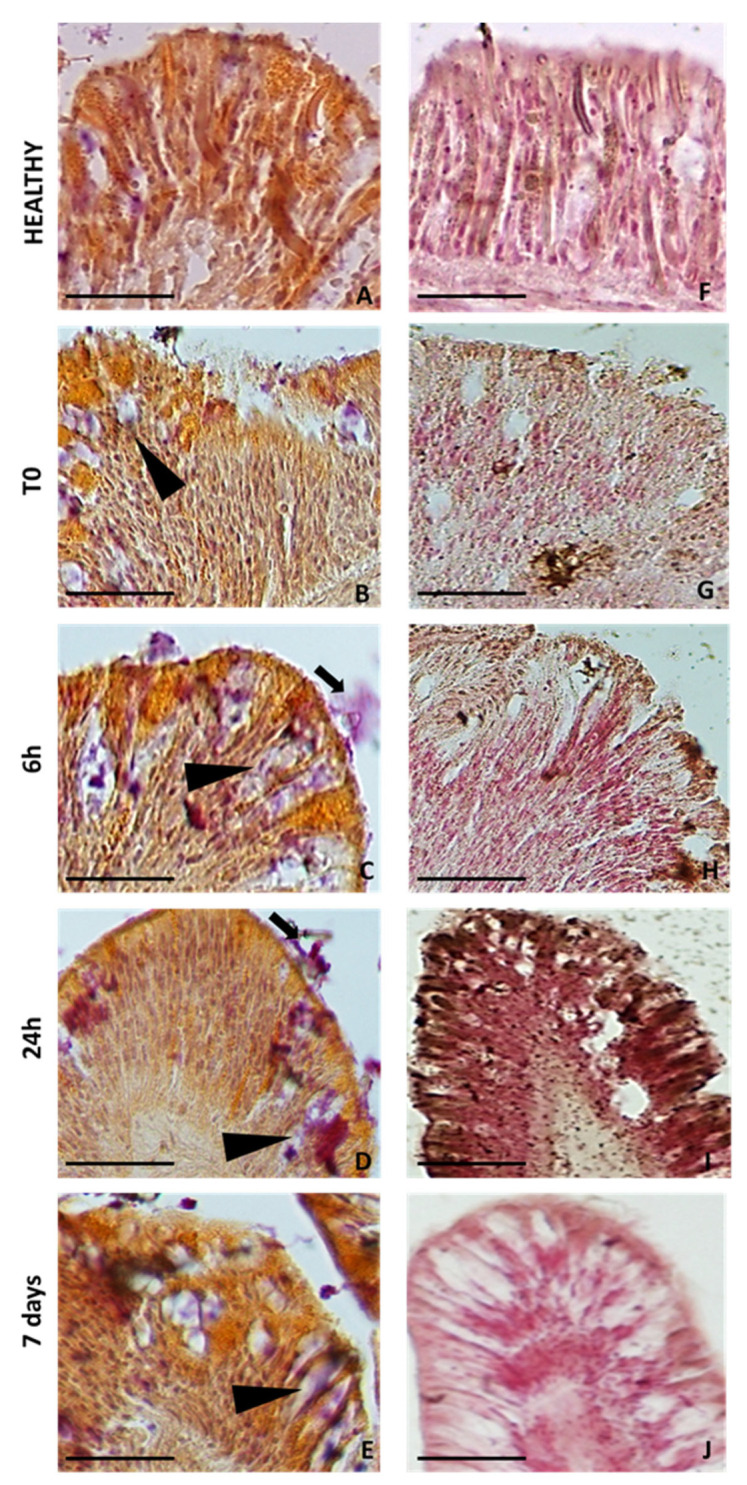
Tissue sections stained with mucicarmine (**A**–**E**) and Fontana–Masson stain (**F**–**J**). Mucus is stained in fuchsia and is localised in mucous cells (arrowheads) and on the surface (arrow) at the edge of the lesion. Melanin-containing granular cells are stained in black. Bar in (**A**–**J**), 25 µm.

**Figure 8 ijms-24-08860-f008:**
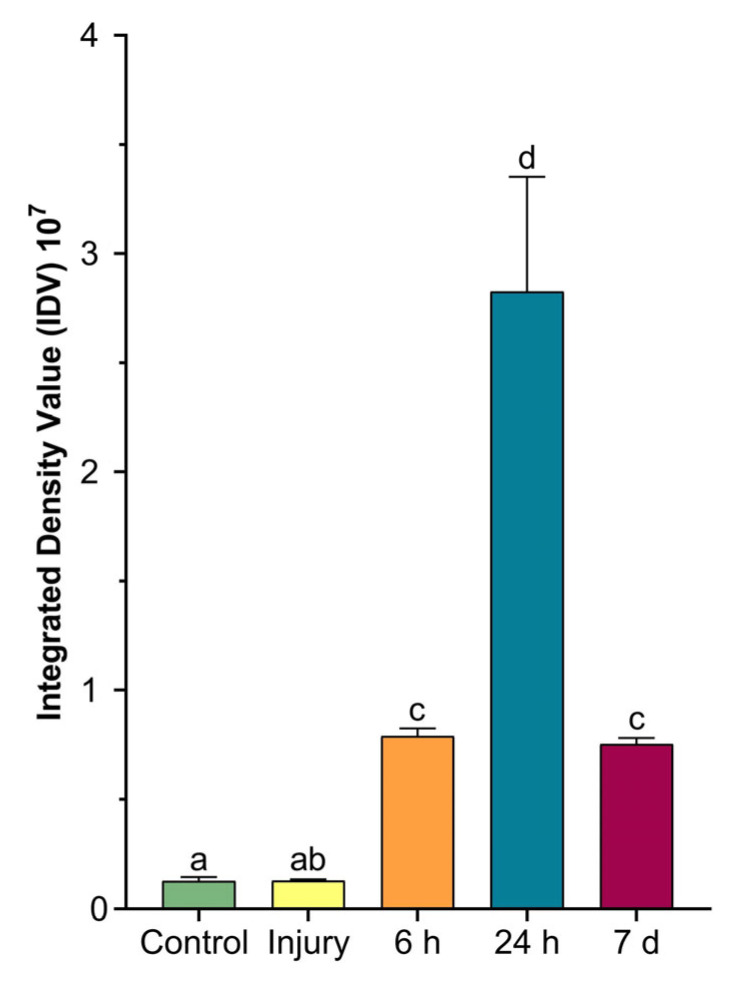
The graph shows the densitometric analysis calculated on *A. viridis* sections stained with Fontana–Masson. Here, the integrated value of density (IDV) represents the quantification of colour intensity at the different time points. Data were tested for ANOVA assumptions, and differences between groups are shown using a one-way ANOVA as mean ± standard deviation (SD). Differences between means were considered significant for *p* < 0.05. Letters (a–d) indicate differences between treatments.

## Data Availability

The authors declare that the data supporting the findings of this study are available within the paper and its Appendix A. The raw data are available from the corresponding author upon reasonable request.

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
