# Peer review of "Step-by-Step Regeneration of Tentacles after Injury in Anemonia viridis—Morphological and Structural Cell Analyses"

_ijms, 2023, doi:10.3390/ijms24108860_

Round 1
Reviewer 1 Report
In the research article by Claudia et al author have reported the wound healing and regeneration events in basal metazoan- Anemonia viridis for the first time. The author performed histological investigation to compare difference between injured and healthy tissue at 0h, 6h, 24h and 7 days. The author also claims that their study is the first to perform Positron Emission Tomography in order to investigate the regeneration event in a longer period time (0 h, 24 h, 14 days after tentacles cut) in invertebrates. They also did Fontana-Masson staining at 24hr after tentacles were cut and reported increase in the melanin-like containing cells and fibroblast-like cells differentiated by amoebocyte at early stage of inflammation and regeneration. Overall, the findings are unique and will be valuable to the readers of IJMS journal, however, there are minor concerns that need to be addressed for research article to be considered for publication.
Comment: Author mentions in the abstract that their results indicate that such processes during wound healing and regeneration are highly conserved in different phyla, however, author has not performed any experiments in the paper that compares such phenomena across phyla. Therefore, I will recommend that the Author mentions it only in the discussion to draw a comparison. Author can also cite some more articles from the past that have discussed in detail the role of evolutionarily conserved methodologies in regeneration among different phyla.
Author Response
We thank the reviewer for the comment. We have reshaped the abstract and quoted more articles on regeneration as suggested.
Reviewer 2 Report
Overview of the manuscript
The manuscript is focuses on the study of regeneration steps after injury caused by tentacles amputation in soft coral Anemonia viridis. The authors investigated the regenerative process by analysing the morphological changes in the amputated structure through the use of classical histololy, TEM and epifluorescence PET. The authors aims to provide elucidations on wound-healing and regeneration events in basal metazoan, focusing on the characterization of immune cells and their role.
GENERAL COMMENT
The work is interesting and well focused on the topic of the work. The detailed documentation step by step of morphological changes during regeneraion of the amputed tentacle is appreciable. The TEM images illustrate very clearly the event discussed in the manuscript and toghether PET methodologies give the manuscript a valid support. In some points the manuscript should be improved in terms of presentation.
SPECIFIC COMMENTS
Abstract
You indicate the “focusing on the characterization of immune cells and their role” but I do not find adequate indications on this topic in Discussion Section
Pag. 1, line 29-30: the reference to “biothechnological studies” remains generic and not defined. This concept does not find indications in the text of the work. Delete it.
Results
Fig. 2: E,F,G are not indicated in the text. Some indications are not clear (for example: “in yellow = starch granule”). Check the indications.
Pag. 2, paragraph 2.2: you should insert pictures to support the morphometric results
Discussion
Pag. 10, line 288: You previous work? Indicate reference.
Pag. 11, line 335: inflammatory pathways are not a detail of your study. Give references or delete the sentence.
In abstract you indicate the characterization of immune cells and their role. But in discussion I can not find an adequate comment on this issue. What cellular elements are involved in the immune cells response?
Materials and Methods
Pag. 12, line 363-368: the distribution of the animals in the several experimental groups is not clear. Please reformulate the sentence.
Pag. 12, line 366: why do you indicate “Treatment”? What treatment have you done? Explain better or correct.
Pag. 12, line 405: what does “integrated density values” means? Explain with more details.
nihil
Author Response
Reviewer 2
We want to thank the reviewer for his helpful comments.
Overview of the manuscript
The manuscript is focuses on the study of regeneration steps after injury caused by tentacles amputation in soft coral Anemonia viridis. The authors investigated the regenerative process by analysing the morphological changes in the amputated structure through the use of classical histololy, TEM and epifluorescence PET. The authors aims to provide elucidations on wound-healing and regeneration events in basal metazoan, focusing on the characterization of immune cells and their role.
GENERAL COMMENT
The work is interesting and well focused on the topic of the work. The detailed documentation step by step of morphological changes during regeneraion of the amputed tentacle is appreciable. The TEM images illustrate very clearly the event discussed in the manuscript and toghether PET methodologies give the manuscript a valid support. In some points the manuscript should be improved in terms of presentation.
SPECIFIC COMMENTS
Abstract
You indicate the “focusing on the characterization of immune cells and their role” but I do not find adequate indications on this topic in Discussion Section
Pag. 1, line 29-30: the reference to “biothechnological studies” remains generic and not defined. This concept does not find indications in the text of the work. Delete it.
-We have modified as suggested.
Results
Fig. 2: E,F,G are not indicated in the text. Some indications are not clear (for example: “in yellow = starch granule”). Check the indications.
- We have improved text and image as suggested.
Pag. 2, paragraph 2.2: you should insert pictures to support the morphometric results
- We added an image as an example of morphometric results in the additional files.
Discussion
Pag. 10, line 288: You previous work? Indicate reference.
-Reference added.
Pag. 11, line 335: inflammatory pathways are not a detail of your study. Give references or delete the sentence.
-Sentence removed.
In abstract you indicate the characterization of immune cells and their role. But in discussion I cannot find an adequate comment on this issue. What cellular elements are involved in the immune cells response?
-We thank the reviewer for the comment. Probably at some points in the discussion we had to clarify better that the cells evidenced, such as granulocytes, glandular cells or mucocytes are also cells involved in immunity since they can secrete active compounds or mucus, thus performing an activity of defense against possible pathogens and activating pathways capable of restoring the normal homeostasis of the organism. We have clarified it in the discussion adding also new references.
Materials and Methods
Pag. 12, line 363-368: the distribution of the animals in the several experimental groups is not clear. Please reformulate the sentence.
-Sentence reformulated.
Pag. 12, line 366: why do you indicate “Treatment”? What treatment have you done? Explain better or correct.
-We have modified the sentence.
Pag. 12, line 405: what does “integrated density values” means? Explain with more details.
-We have added an explanation.
Reviewer 3 Report
This manuscript is the first to use Positron Emission Tomography in invertebrates to study regeneration events over extended time periods. This is very meaningful and provides a new perspective for the regeneration research of invertebrates.
Author Response
We greatly appreciate the reviewer’s comment and thank him for the time he devoted to reviewing the work.